# Bio-Fertilizers Reduced the Need for Mineral Fertilizers in Soilless-Grown Capia Pepper

Hayriye Yildiz Dasgan [1,*], Mehmet Yilmaz [1], Sultan Dere [1,2], Boran Ikiz [1] and Nazim S. Gruda [3,*]

1   Department of Horticulture, Faculty of Agriculture, University of Cukurova, Adana 01330, Turkey
2   Department of Horticulture, Faculty of Agriculture, University of Siirt, Siirt 56100, Turkey
3   Institute of Plant Sciences and Resource Conservation, Division of Horticultural Sciences, University of Bonn, 53121 Bonn, Germany
*   Correspondence: dasgan@cu.edu.tr (H.Y.D.); ngruda@uni-bonn.de (N.S.G.)

**Abstract:** Soilless cultivation is extensively used in the greenhouse industry. Recently, hydroponic cultivation of capia pepper has become popular among growers. Capia pepper is harvested at the red maturity stage, and intensive mineral fertilizers are usually used for soilless cultivation. This study was performed in a greenhouse during spring under Mediterranean climatic conditions. The effects of bacteria and mycorrhiza on capia pepper plant growth, yield, fruit quality, and nutrition were investigated. Furthermore, the synergistic effects of these two bio-fertilizers were investigated. Our objective was to replace 20% of mineral fertilizers with bio-fertilizers in a soilless culture system. The use of 80% mineral fertilizers, in combination with mycorrhiza and bacteria, provided a 32.4% higher yield than the control (100% mineral fertilizer without bio-fertilizers). Moreover, the concentrations of N, P, K, Ca, Mg, Fe, Mn, Zn, and Cu in the leaves of pepper plants fed with the reduced mineral fertilizers combined with bio-fertilizers were higher than that of the control. In addition, fruit parameters, such as fruit weight, diameter, volume, the electric conductivity of the fruit juice, and total soluble solids, were significantly higher in this treatment compared to the control. Using 80% mineral fertilizer with only bacteria provided a 24.2% higher yield than the control. In conclusion, mineral fertilizers were successfully reduced by 20% using bacteria and mycorrhiza. These results provide an eco-friendly approach to a sustainable environment.

**Keywords:** bacteria; *Capsicum annuum* L.; coco pith; mycorrhiza; synergistic effects; yield

## 1. Introduction

Intense monoculture vegetable growing in conventional soil greenhouses increases the risk of disease and pest outbreaks. Consequently, high amounts of pesticides and herbicides are needed, increasing environmental pollution. On the other hand, this cultivation method can reduce certain nutrients' availability, leading to soil exhaustion. Our endeavors to compensate for the situation could, in turn, increase the level of unbalanced nutrients and negatively affects soil fertility. For instance, long-term chili monoculture generated significant changes in soil nutrients, aggregates and enzymes [1]. All these factors severely limit crop productivity over the years.

In recent years, the gradual decrease in agricultural lands, the effects of climate change, soil-borne problems, food security, and environmental issues have increased the tendency towards soilless cultivation as controlled agriculture [2,3]. A soilless culture system (SCS) is considered one of the most promising approaches, combining increased production without damaging its supporting ecosystem [3]. Likewise, soilless culture systems have been adopted in many countries for a long time. Plant nutrition management is a crucial factor in the success of soilless cultivation systems. The right nutrient solution will increase the complete fulfillment of plants' requirements for optimal growth and development. However, although soilless cultivation in the modern greenhouse business is the favorite

choice among growers, it brings intensive nutrition to the plants through mineral fertilizers. Coco pith is an organic substrate produced from the wastes of coco fruits. It has a high water-holding capacity, cation-exchange capacity, aeration, a light bulk density, an ideal pH and EC, and is free of pathogens. These properties have made it one of the most preferred soilless growing media in the greenhouse industry in recent years [3]. Therefore, coco pith was chosen for capia pepper cultivation in this study.

In recent years, improvements in beneficial microorganisms have increased the tendency to use bio-fertilizers as valuable tools in sustainable agriculture. Unlike in soil, in hydroponic growing soilless systems there are no beneficial microorganisms in the root environment, and plants cannot benefit from these microorganisms. Therefore, bio-fertilizers such as mycorrhiza fungi, bacteria, and microalgae are commonly adapted to soilless growing systems [4–7]. Bio-fertilizers are effective strains of microorganisms that help crop plants' nutrition by enhancing growth, yield, and crop quality in soilless culture systems [8]. At the same time, they provide environmentally friendly agriculture by reducing the use of mineral fertilizers. Inoculation of arbuscular mycorrhizal fungi (AMF) and plant-growth-promoting rhizobacteria (PGPR) become an alternative solution to increase the efficiency of nutrient usage as well as absorption of nutrients by plants [5,9]. The use of bio-fertilizers in the soilless growing of several vegetable species was previously reported in related literature [10–13].

Similarly, studies also acknowledged the beneficial effects of bio-fertilizers on pH and electrical conductivity (EC) of the nutrient solution regulations, chelator secretion in the rootzone, increased uptake of nutrients, plant growth, yield, and crop quality [7,14]. In an SCS, the nutrient solution that consists of mineral fertilizers is used in every irrigation. An open system means the nutrient solution is not recyclable, and the excess drainage solution (about 20–25% of the applied nutrient solution) can pollute the environment and groundwater resources. In the greenhouse sector, it is noteworthy that a significant majority of SCSs produce crops in an open system. For this reason, less mineral fertilizer means environmental protection. Bio-fertilizers improve, in addition, the pH and EC of the nutrient solution in the root zone [7]. Therefore, they provide a high intake of nutrients from plants. In addition, biofertilizers decrease nitrate and increase the antioxidants and minerals of vegetables [13,15]. Hence, biofertilizers provide positive contributions to human nutrition. However, studies on bio-fertilizers used in reducing mineral fertilizers are quite limited. On the other hand, farmers' and consumers' awareness of environmental and ecological concepts has recently increased. As a result, some farmers who want to produce according to a sustainable model prefer to use organic fertilizers and/or bio-fertilizers in soilless cultivation systems [16].

Capia (*Capsicum annuum* L.) is a type of pepper with a long conical shape, red color in maturity, and a sweet taste; it is rich in vitamins C and A, folic acid, potassium, mineral substances, phenolic compounds, carotene, and antioxidant compounds [17,18]. It has a dense juicy pulp with a rich aroma in the red maturity stage. Capia pepper fruit at the red ripeness stage has the following morphological properties: 80–125 g weight, 15–20 cm length, 40–55 mm width, 150–250 $cm^3$ volume, and 3.5–4.0 mm flesh thickness. Capia pepper seeds germinate in 7–10 days. The seedlings grow in around 30–40 days. The plant grows best between 22–25 °C during the day and 15–18 °C at night. An increase in daytime temperature to 28 °C accelerates red ripening. Red ripe capia fruits are harvested in approximately 90 days from seedling planting. Capia pepper, very popular in the Balkan countries and Turkey, is commonly used for pepper paste, pepper juice, pickles, frozen products, frying, and peppery sauce in summer field production [18].

On the contrary, greenhouse capia is used for table consumption during the cold winter season. The capia pepper's soilless cultivation in the greenhouse has recently become popular among growers. Approximately 3,018,775 tons of pepper are produced in Turkey, and 49% of this is capia pepper [19]. The consumer preference in Turkey is primarily capia pepper. Recently, capia cultivation has increased with soilless techniques in

the greenhouse. Greenhouse producers search for environmentally friendly fertilizers that increase product yield and quality.

To our knowledge, reducing mineral fertilizers and substituting them with bio-fertilizers has not been previously investigated in SCSs for capia pepper. Therefore, this study aimed to reduce the intensive use of mineral fertilizers by inoculating with beneficial microorganisms such as AMF and PGPR. Moreover, bio-fertilizers' effects on plant growth, yield and fruit quality of soilless-grown capia pepper were investigated in this study.

## 2. Materials and Methods

### 2.1. Plant and Bio-Fertilizer Materials and Experimental Conditions

This study was conducted in a glasshouse at 36°59′ N, 35°18′ E, and 23 m above the Mediterranean Sea level in the early spring growing season (February-July). Climatic conditions inside the glasshouse were 23–25 °C during the day and 15–20 °C at night, with 50–60% relative humidity and natural sunlight conditions. Trademarked Lale $F_1$ capia pepper of the Istanbul Tarim company was used. We used coco pith slabs as a soilless cultivation medium, cultivating four plants in every slab, with four slabs in each replication. The size of the coco pith slab was 100 cm long, 20 cm wide, and 10 cm deep. For the randomized complete block experimental design with seven treatments and four replicates, 16 plants were used in each replicate. Pepper seedlings 35 days old were transplanted into the coco pith slabs 25 cm above the row and 80 cm between the rows Figure S1.

The mycorrhiza bio-fertilizer Endo Roots Soluble (ERS), a cocktail from nine different mycorrhiza species: Glomus intraradices, Glomus aggregatum, Glomus mosseae, Glomus clarum, Glomus monosporus, Glomus deserticola, Glomus brasilianum, Glomus etunicatum, and Gigaspora margarita, was used. The liquid bacteria Medbio bio-fertilizer used in the experiment contained four different bacteria species: Bacillus subtilis ($1 \times 10^9$), Bacillus licheniformis ($2 \times 10^6$), Bacillus megaterium ($1 \times 10^9$) and Pseudomonas putita ($1 \times 10^{10}$). Pepper seedlings were inoculated only once during transplanting, with approximately 2000 mycorrhizae spores per plant; with the growth of the root system, mycorrhiza spores multiply in a symbiosis relationship. Meanwhile, bacteria were applied every ten days to the roots during growing. PGPR was applied at 50 mL per plant from the 1 mL Metbio in 1 L nutrient solution. Repeated use of PGPR can stabilize the number of bacteria in the rootzone. Preliminary trials were carried out to determine the bacteria application method. Very successful results were obtained from these preliminary trials. The method used did not ever lead to uneven application or other causes of error. Soilless pepper plants, provided with a nutrient solution with 100% mineral fertilizers, served as a control (Table 1) [20]. Moreover, we substituted 20% and 40% of the mineral fertilizers with mycorrhiza, bacteria, and their combination. In the study, seven treatments were applied.

**Table 1.** The nutrient solution used in the control treatment (100% mineral fertilization) (mg L$^{-1}$).

| N | P | K | Ca | Mg | Fe | Mn | Zn | B | Cu | Mo |
|---|---|---|---|---|---|---|---|---|---|---|
| 100–239 | 40–81 | 96–370 | 150–250 | 50–92 | 5–10 | 1.97 | 0.25 | 0.7 | 0.07 | 0.05 |

1. Common nutrient solution, 100% mineral fertilization (as control) (Table 1),
2. 60% mineral fertilization (MF) + PGPR,
3. 60% MF + AMF
4. 60% MF + PGPR + AMF
5. 80% MF + PGPR
6. 80% MF + AMF
7. 80% MF + PGPR + AMF

## 2.2. Nutrient Solution and Irrigation

The amount of nutrient solution applied to the plants was determined based on the daily drainage ratio (DR) from the base of the coco pith slabs. The drainage ratio was approximately 20% $\pm$ 5 [21]. The pH and EC of the nutrient solution during the cultivation period were maintained within the range of 6.0–6.5 and 2.0–2.8 dS m$^{-1}$, respectively.

$$DR = \text{drainage solution (mL)} \div \text{applied nutrient solution (mL)} \times 100$$

## 2.3. Parameters Examined in the Experiment

Plant growth parameters, such as plant height, stem diameter and the number of branches, were measured 80 days after seedling transplanting (DAT) (Table 2). In addition, shoot and leaf fresh weights and leaf area per plant were recorded at 164 DAT at the end of the experiment. The leaf area was determined by a leaf area meter (Li-3100, LICOR, Lincoln, NE, USA) and indicated as cm$^2$ plant$^{-1}$. Ten plants per plot were used for the measurements.

**Table 2.** Effects of the bio-fertilizers on plant height, diameter and branches, 80 DAT.

| Treatments | Plant Height (cm) | Stem Diameter (mm) | Number of Branches |
|---|---|---|---|
| 100%MF | 81.08 b | 10.66 | 6.42 |
| 60%MF + PGPR | 78.75 b | 9.61 | 5.75 |
| 60%MF + AMF | 66.50 c | 9.00 | 5.67 |
| 60%MF + PGPR + AMF | 83.25 b | 9.75 | 5.33 |
| 80%MF + PGPR | 90.83 a | 10.04 | 5.20 |
| 80%MF + AMF | 94.92 a | 10.03 | 5.33 |
| 80%MF+ PGPR + AMF | 98.13 a | 10.09 | 4.92 |
| LSD$_{0.05}$ | 7.398 | NS | NS |
| P | <0.0001 | 0.1620 | 0.5873 |

DAT: days after transplanting; MF: mineral fertilizer; PGPR: plant-growth-promoting rhizobacteria; AMF: arbuscular mycorrhizal fungi; NS: not significant. Different letters within a column indicate significant differences.

Pepper fruits were harvested weekly when they reached the red maturity stage (Figure 1). The cumulative yield of pepper fruit is expressed as kg m$^{-2}$ for the total harvest. Red ripe pepper fruit sampling, 15 fruits per replication, was used for fruit quality measurements. The pH, EC, total soluble solids (TSS), and titratable acidity were measured in the capia pepper fruit.

### 2.3.1. Determination of Leaf Potassium (K), Calcium (Ca), Magnesium (Mg), Iron (Fe), Zinc (Zn), Manganese (Mn), and Copper (Cu) by Atomic Absorption Spectrophotometry

Leaf samples, 20 fully mature leaves of 10 plants per replicate, were collected at 80 DAT for mineral nutrient analysis. Leaves were dried in a forced-air oven at 65 °C for 48 h and ground through a 40-mesh sieve for elemental analysis [21]. The samples were dry-ashed in a muffle furnace at 550 °C for six hours. The ash was then dissolved in 0.1 M hydrochloric acid (HCl). K, Ca, Mg, Fe, Mn, Zn, and Cu concentrations were determined using an atomic absorption spectrophotometer [22].

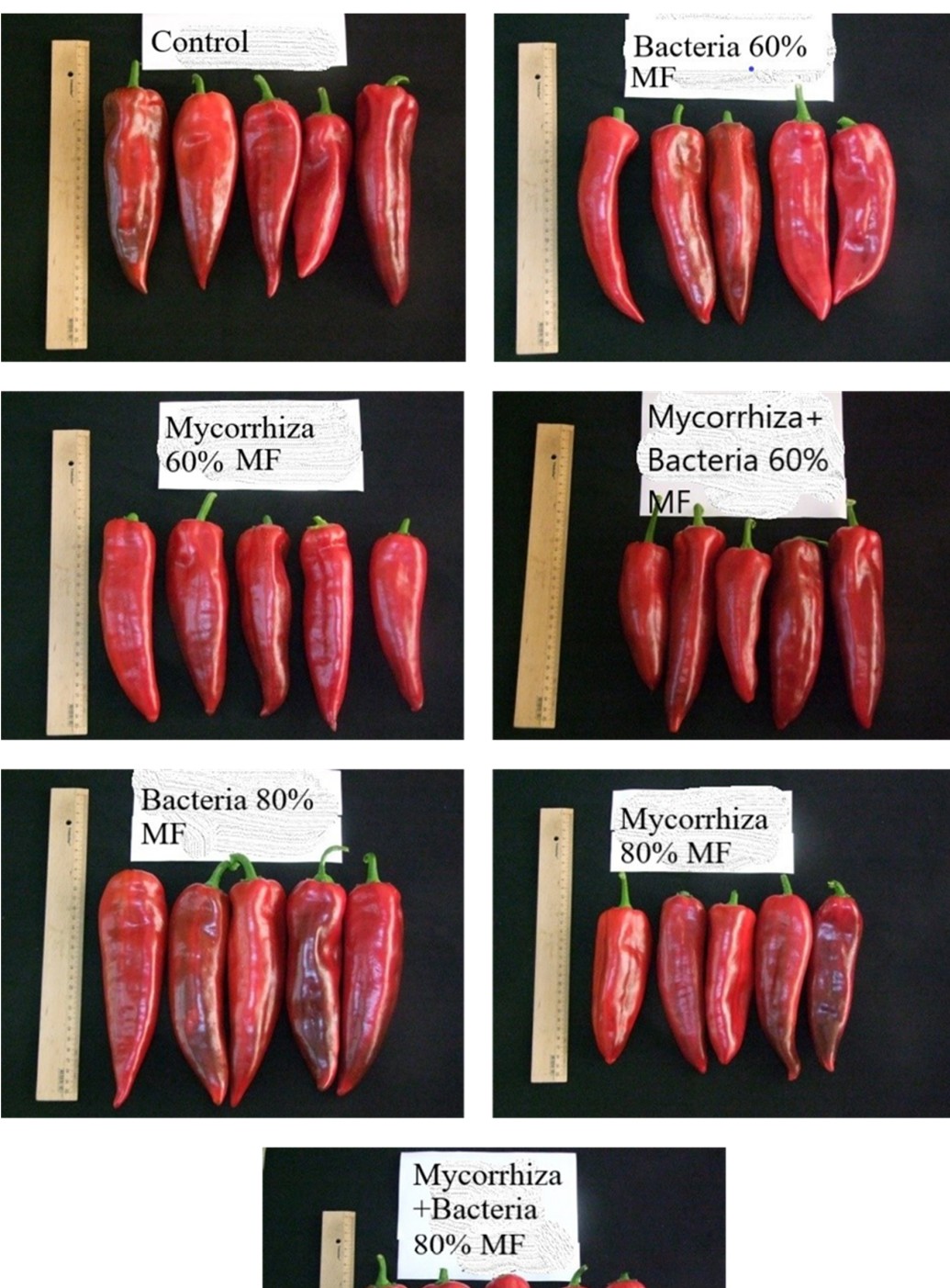

**Figure 1.** Soilless-grown capia peppers were harvested weekly when they reached the red maturity stage. MF: mineral fertilizer.

2.3.2. Determination of Leaf Total Nitrogen (N) by the Kjeldahl Method

Dry-ground leaf samples weighing 1 gm were weighed out; 5 mL of concentrated $H_2SO_4$ and a selenium tablet were placed on them; there were burned in the combustion unit of a Kjeldahl apparatus at 400 °C for 1 h until the color turned pale. Then, distillation was performed with 28% NaOH in a Kjeldahl tube distillation apparatus. Boric acid and the indicator solution were added to the ammonia released during distillation, and then, titration was performed with 0.01 N HCl. The total nitrogen of the leaf was calculated with the amount of HCl consumed in the titration (modified from [22]).

2.3.3. Determination of Leaf Phosphorus (P) by the Barton Method

The dry-ashed, furnaced and dissolved leaf samples (as mentioned above) were reacted with Barton's solution. The phosphorus concentration was determined at a wavelength of 430 nm in the spectrophotometer (modified from [22]).

*2.4. Statistical Analysis*

Data were analyzed using one-way analysis of variance (ANOVA) with the SAS-JMP/7 statistical program. The averages of the treatments were compared with the least significant difference (LSD) test at $p \leq 0.05$ level.

**3. Results**

*3.1. Effects of Bio-Fertilizers on Plant Growth*

The synergistic effect of mycorrhiza and bacteria with 80% mineral fertilizer increased pepper plant height in our experiment. They were approximately 21% taller than the control. The tallest pepper plants were grown by using both bio-fertilizers (Table 2). Pepper plants grown in 60% MF + PGPR + AMF were as tall as the control plants. There was no statistically significant difference between treatments for plant stem diameter and number of branches (Table 2). Lucas et al. [23] reported a positive enhancing effect of the beneficial bacteria *Bacillus licheniformis* on tomato and pepper plant height and stem thickness. Moreover, *Bacillus licheniformis* increased auxins and gibberellins in the pepper and tomato plants.

AMF are beneficial to plants by mobilizing nutrients in the root zone, production of siderophores, improving nutrient and water use efficiencies, promoting nutrient uptake of roots, protecting plants from pathogens, and increasing plants' tolerance for abiotic stresses [24]. PGPR are also beneficial to plants by increasing photosynthesis and stimulating the production of phytohormones (indole acetic acid-IAA, cytokinin, gibberellin), secondary metabolic products such as vitamins, and amino acids [25,26]. At the end of the experiment, the heaviest shoot and leaf weights after the control were 545.0 and 262.3 g plant$^{-1}$, using bacteria and mycorrhiza with 80% mineral fertilizer (Table 3). Similarly, a positive effect of bacteria on tomato and pepper leaf area was found [23]. However, the combination of bacteria and mycorrhiza induced a synergistic effect in our experiment. The photosynthetic performance of the plants largely determines biomass. Kandiannan et al. [27] investigated two bacteria and one mycorrhiza, applied in single, double, and triple combinations to the black pepper plant (*Piper nigrum*) grown in containers. The double and triple combinations significantly increased plant height, leaf area, and biomass production compared to the control plants.

**Table 3.** Effects of the bio-fertilizers on shoot and leaf weights 164 DAT at the end of the cultivation.

| Treatments | Shoot Fresh Weight (g Plant$^{-1}$) | Leaf Fresh Weight (g Plant$^{-1}$) |
|---|---|---|
| 100%MF | 635.2 a | 297.5 a |
| 60%MF + PGPR | 377.4 cd | 185.3 de |
| 60%MF + AMF | 308.2 d | 157.7 e |
| 60%MF + PGPR + AMF | 412.8 bcd | 206.3 cd |
| 80%MF + PGPR | 530.6 abc | 258.0 b |
| 80%MF + AMF | 516.8 abc | 243.3 bc |
| 80%MF+ PGPR + AMF | 545.0 ab | 262.3 ab |
| LSD$_{0.05}$ | 156.484 | 34.008 |
| *P* | 0.0053 | <0.0001 |

DAT: days after transplanting; MF: mineral fertilizer; PGPR: plant growth promoting rhizobacteria, AMF: arbuscular mycorrhizal fungi. Different letters within a column indicate significant differences.

The leaf area of pepper plants increased by using biofertilizers. The leaf is the major photosynthetic apparatus of plants. A synergistic effect of microorganisms was observed for this parameter. AMF, PGPR and 80% mineral fertilizer resulted in a leaf area close to the control (Figure 2). The increase in leaf area might be due to increased nutrient availability due to the production of phytohormones. This, in turn, caused an enhancement in plant growth and fruit yield. As an efficient photosynthetic organ, the leaf area most likely induced the building of more plant carbohydrates [13].

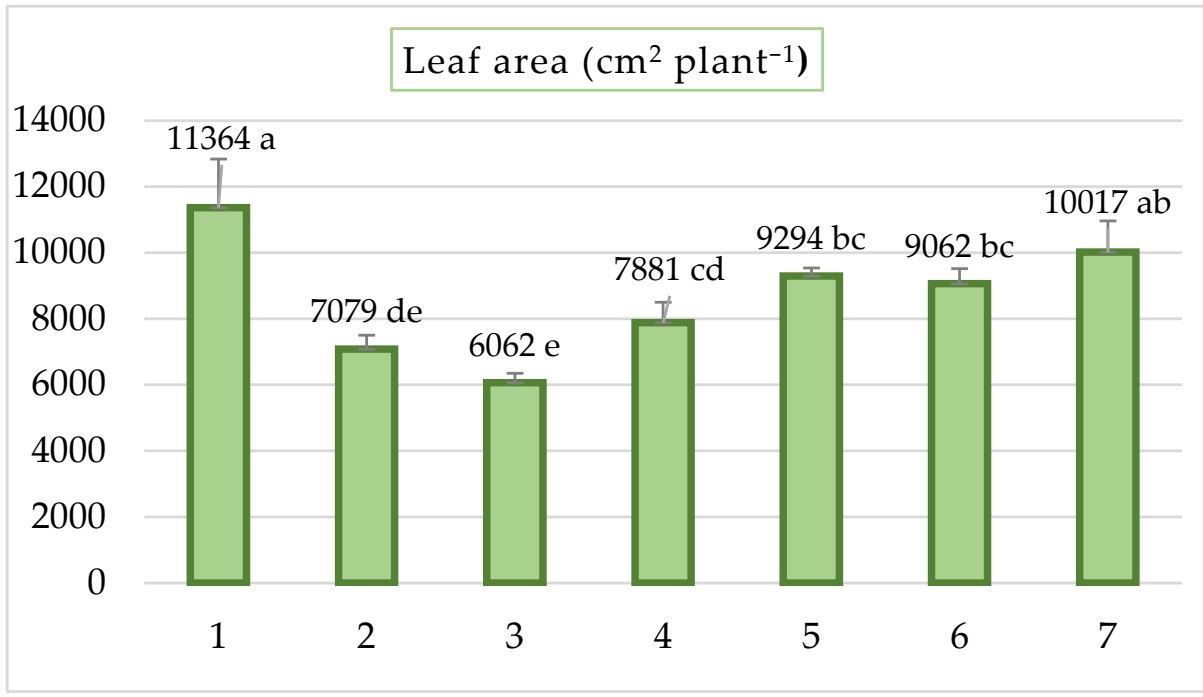

**Figure 2.** Effects of biofertilizers on pepper plant leaf area 164 DAT at the end of cultivation. DAT: day after transplanting. 1:100% MF, 2:60% MF + PGPR, 3:60% MF + AMF, 4:60% MF + PGPR + AMF, 5: 80% MF + PGPR, 6:80% MF + AMF, 7:80% MF + PGPR + AMF. MF: mineral fertilizer; PGPR: plant-growth-promoting rhizobacteria; AMF: arbuscular mycorrhizal fungi. Different letters within a column indicate significant differences, LSD$_{0.05}$:1413, *p* < 0.0001.

### 3.2. Effects of Bio-Fertilizers on Total Fruit Yield and Fruit Number

Total pepper fruit yield ranged from 3.30 to 5.80 kg m$^{-2}$. The application of bacteria and mycorrhiza to 80% mineral fertilizer induced the highest yield, 32.4% higher than the control (4.38 kg m$^{-2}$). The second- and third-highest yields were 24.2% and 11.2% for the 80% MF + PGPR and 80% MF + AMF treatments, respectively (Figure 3). In bio-fertilized plants with the consortium of PGPR + AMF, better plant growth and biomass production could promote photosynthesis more effectively. Therefore, accumulated supply facilitates fruit development and contributes to a higher total yield. According to Dere et al. [2], bacteria and mycorrhiza can enhance plant nutrient uptake and, in turn, photosynthesis. The lowest yield was obtained from 60% MF + AMF with 3.30 kg m$^{-2}$ with a 16.4% yield decrease compared to the control. Maboko et al. [28] grew soilless tomato plants at 25% and 50% low nutrient levels with mycorrhiza in heated and unheated tunnels. Mycorrhiza worked more effectively in heated tunnels and increased tomato yields at the reduced nutritional treatments. Perhaps the temperatures in heated tunnels contributed to better-establishing mycorrhiza fungus. Baum et al. [29] reported that mycorrhizal inoculation increased pepper plant growth, fresh biomass, and total yield. Aini et al. [5] found that soilless-grown lettuce associated with PGPR + AMF increased the synthesis of growth-promoting plant hormones, primarily cytokines, which enhances leaf growth. While leaf growth contributes to canopy development, a greater photosynthetically active surface area becomes available, improving plant growth and yield. El-Tohamy et al. [9] reported similar results. Bio-fertilization resulted in higher N, P, and K contents of tomato leaves and higher indole acetic acid, gibberellins, and cytokines. Backer et al. [16] reported that mixing bacteria with mycorrhizal fungi improved corn, tomato, and soybean yields.

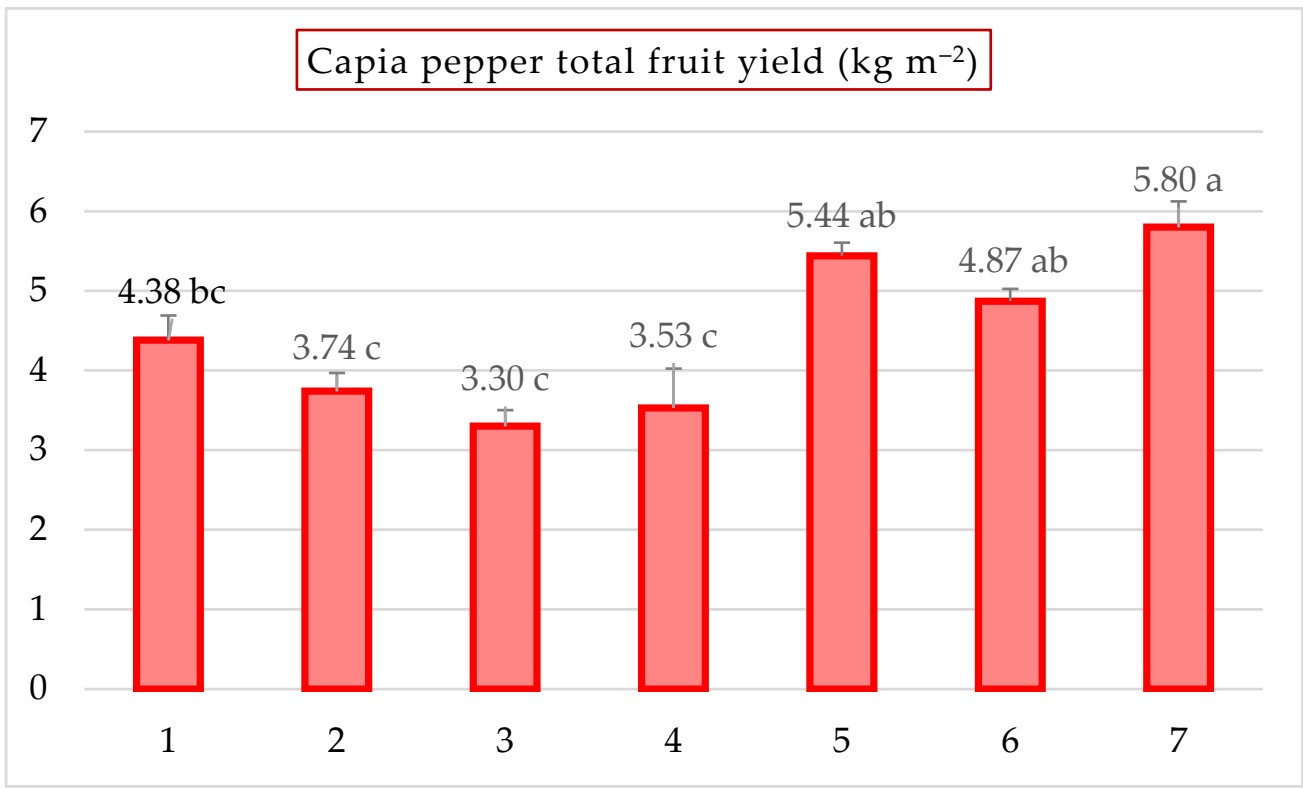

**Figure 3.** Effects of the bio-fertilizers on total fruit yield of soilless-grown capia pepper under reduced mineral fertilizers in the Mediterranean climate in spring greenhouse conditions. 1:100% MF, 2:60% MF + PGPR, 3:60% MF + AMF, 4:60% MF + PGPR + AMF, 5:80% MF + PGPR, 6:80% MF + AMF, 7:80% MF + PGPR + AMF. MF: mineral fertilizer; PGPR: plant-growth-promoting rhizobacteria; AMF: arbuscular mycorrhizal fungi. Different letters within a column indicate significant differences, LSD$_{0.05}$: 1.134, *p*: 0.0023.

The number of pepper fruit harvested per m$^2$ in our experiment varied from 35 to 48 fruit m$^2$. The highest number of fruits was obtained with the treatment using 80% mineral fertilizer combined with bacteria (Figure 4). The presence of 80% mineral fertilizer with bacteria provided better fruit set and development. The number of fruits per unit area of 80% MF + AMF is lower than 80% MF + PGPR and equal to 100% MF (Figure 4). Since the yields of the 80% MF + AMF and 80% MF + PGPR per unit area were in the same significance level and higher than the 100% MF (Figure 3), single-fruit weight of 80% MF + AMF was higher than that of the 80% MF + PGPR and 100% MF (Table 4). It seems that biofertilizer ingenuity induced better plant nutrition and promoted photosynthesis and fruiting.

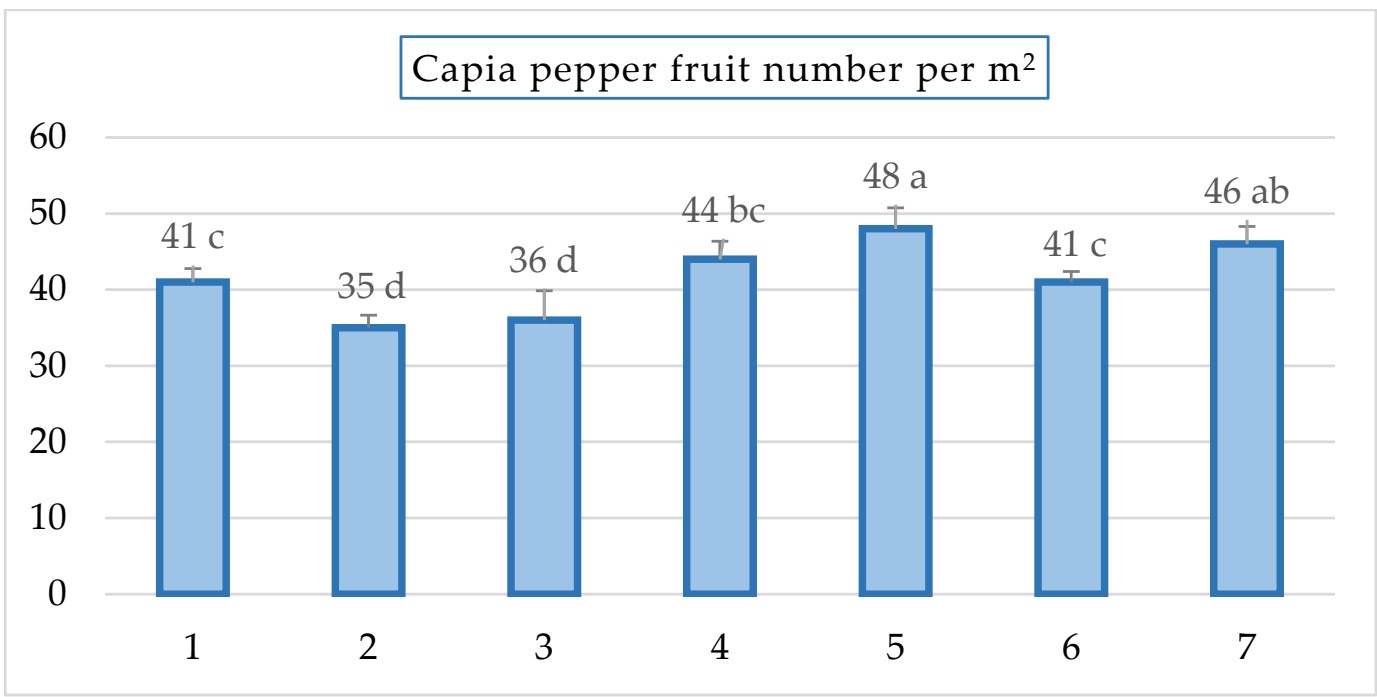

**Figure 4.** Effects of the bio-fertilizers on total fruit number of soilless-grown capia pepper under reduced mineral fertilizers in the Mediterranean climate in spring greenhouse conditions. 1:100% MF, 2:60% MF + PGPR, 3:60% MF + AMF, 4:60% MF + PGPR + AMF, 5:80% MF + PGPR, 6:80% MF + AMF, 7:80% MF + PGPR + AMF. MF: mineral fertilizer; PGPR: plant-growth-promoting rhizobacteria; AMF: arbuscular mycorrhizal fungi. Different letters within a column indicate significant differences, LSD$_{0.05}$: 4.119, *p*: <0.001.

### 3.3. Effects of Bio-Fertilizers on Capia Pepper Fruit Quality Properties

Using soilless culture systems to control nutrients and temperature in the rootzone and managing environmental and agronomic factors can improve product quality [28,30,31]. Dasgan et al. [13] showed that biofertilizers could affect the leaf yield, nitrate amount, and mineral and antioxidant content of basil (*Ocimum basilicum* L.) in a floating culture. In addition, Baum et al. [29] summarized that many research results prove the positive effects of AMF on plant growth, P crop physical and chemical characteristics, and produce quality. Our study's average pepper fruit weight ranged from 91.96 to 125.16 g. The treatment containing 80% mineral fertilizers with mycorrhiza and bacteria produced 19% heavier fruits than the control. As in the case of the total yield, the combined use of bacteria and mycorrhiza showed synergistic effects on fruit growth and physical properties, e.g., firmness and flesh thickness (Table 4). The second-heaviest fruit was from the 80% mineral fertilizers with mycorrhiza treatment (118.86 g), which was 13% heavier than the control. Effects of bio-fertilizers on pepper fruit height and diameter also increasingly affected the fruit weight. Dasgan et al. [15] reported that when the mineral fertilizers were reduced

by 20% and 40%, mycorrhiza, bacteria, and microalgae increased soilless-grown squash size. Although the effects of the treatments on pepper firmness and flesh thickness were insignificant, the effects on fruit volume were remarkable (Table 4). The maximum fruit volume was obtained from the 80% MF + PGPR + AMF treatment, with 246 cm$^3$ (Table 4).

**Table 4.** Effects of the treatments on the physical properties of capia pepper fruit.

| Treatments | Weight (g) | Height (cm) | Diameter (mm) | Volume (cm$^3$) | Firmness (kg cm$^{-3}$) | FLESH Thickness (mm) |
|---|---|---|---|---|---|---|
| 100%MF | 105.11 c | 20.33 a | 49.69 cd | 201 bc | 4.21 | 4.33 |
| 60%MF + PGPR | 109.70 bc | 19.95 a | 51.59 bc | 208 ab | 4.00 | 4.23 |
| 60%MF + AMF | 91.96 d | 19.25 a | 47.68 d | 163 c | 4.13 | 3.93 |
| 60%MF + PGPR + AMF | 81.81 d | 16.78 b | 48.24 d | 168 c | 4.16 | 3.65 |
| 80%MF + PGPR | 112.54 bc | 19.70 a | 55.40 a | 231 ab | 3.81 | 4.03 |
| 80%MF + AMF | 118.86 ab | 19.40 a | 54.88 ab | 230 ab | 4.04 | 4.05 |
| 80%MF+ PGPR + AMF | 125.16 a | 20.73 a | 54.94 a | 246 a | 3.98 | 4.12 |
| LSD$_{0.05}$ | 11.382 | 1.845 | 3.321 | 38.531 | NS | NS |
| *P* | <0.0001 | 0.0075 | 0.0001 | 0.0013 | 0.4228 | 0.3261 |

MF: mineral fertilizer; PGPR: plant-growth-promoting rhizobacteria; AMF: arbuscular mycorrhizal fungi; NS: not significant. Different letters within a column indicate significant difference.

Since mycorrhiza and bacteria increased the production of carbohydrates by enhanced photosynthesis and nutrient uptake, it may also have supported significantly increased electrical conductivity (EC) and total soluble solids of the pepper fruit (Table 5). The effects of the treatments on pH and titratable acidity were similar, although there were no statistically significant differences among the treatments (Table 5). Maboko et al. [28] found that inoculating mycorrhiza into soilless-grown tomatoes increased fruit quality, especially total soluble solids and total dry matter. Baum et al. [29] reported that mycorrhiza increased ascorbic acid in chili pepper, beta carotene in potatoes, and carotenoids, phenolics, anthocyanins, chlorophyll and some mineral nutrients in lettuce. In conclusion, bio-fertilizer affects the product quality of vegetables. Michałojć et al. [30] investigated the effect of mycorrhiza in two nutrient solutions with EC 2600 and 1900 µS cm$^{-1}$ on the quality of soilless-grown tomatoes. Tomato fruits produced with mycorrhiza contained significantly more total soluble solids than non-mycorrhizal ones.

**Table 5.** Effects of the treatments on chemical properties of capia pepper fruit.

| Treatments | EC (µS cm$^{-1}$) | pH | TSS (%) | TA (%) |
|---|---|---|---|---|
| 100%MF | 2070 b | 5.27 | 5.90 d | 1.35 |
| 60%MF + PGPR | 2642 a | 5.01 | 7.00 a | 1.79 |
| 60%MF + AMF | 2671 a | 5.04 | 7.10 a | 1.67 |
| 60%MF + PGPR + AMF | 2487 a | 5.10 | 6.93 ab | 1.78 |
| 80%MF + PGPR | 2542 a | 5.16 | 6.10 cd | 1.61 |
| 80%MF + AMF | 2577 a | 5.00 | 6.23 bcd | 1.71 |
| 80%MF+ PGPR + AMF | 2445 a | 5.11 | 6.85 ab | 1.61 |
| LSD$_{0.05}$ | 288.059 | NS | 0.326 | NS |
| *p* | 0.007 | 0.193 | 0.0222 | 0.242 |

MF: mineral fertilizer; PGPR: plant-growth-promoting rhizobacteria; AMF: arbuscular mycorrhizal fungi; NS: not significant; EC: electrical conductivity; TSS: total soluble solids; TA: titratable acidity. Different letters within a column indicate significant differences.

A sensory panel test was conducted by untrained amateur groups and reports showed that there was no significant difference among the treatments for the taste of the pepper such as spiciness, texture and flavor (unpublished data).

### 3.4. Effects of Bio-Fertilizers on Mineral Nutrient Concentration

Several studies reported that bacteria and mycorrhizae contribute to plant growth by increasing mineral nutrient uptake [25,30,32,33]. The concentrations of macro elements N, P, K, Ca, Mg, and microelements Fe, Mn, Zn, and Cu in the leaves of pepper plants provided with bio-fertilizers were higher than that of the leaves of the control treatment supplied with 100% mineral fertilizers (Tables 6 and 7). The bacteria and mycorrhiza used in this experiment improved plant nutrition by supplying and facilitating nutrient uptake. According to the pepper plant tissue analysis and interpretation of Hochmuth et al. [34], the mineral nutrition status of the plants fed with the biofertilizers was determined to be sufficient.

**Table 6.** Effects of the bio-fertilizers on pepper leaf macronutrient concentrations (%).

| Treatments | N | P | K | Ca | Mg |
|---|---|---|---|---|---|
| 100%MF | 3.46 c | 0.37 c | 4.94 d | 1.28 d | 1.10 c |
| 60%MF + PGPR | 3.73 c | 0.40 c | 5.04 cd | 1.94 ab | 1.16 abc |
| 60%MF + AMF | 3.98 bc | 0.61 a | 5.32 bc | 1.88 abc | 1.21 ab |
| 60%MF + PGPR + AMF | 4.34 b | 0.52 b | 5.51 b | 1.64 bc | 1.11 bc |
| 80%MF + PGPR | 5.65 a | 0.49 b | 5.61 ab | 1.59 c | 1.22 a |
| 80%MF + AMF | 5.62 a | 0.48 b | 5.88 a | 2.06 a | 1.26 a |
| 80%MF+ PGPR + AMF | 5.57 a | 0.42 c | 5.89 a | 1.88 abc | 1.23 a |
| $LSD_{0.05}$ | 0.536 | 0.0496 | 0.341 | 0.296 | 0.104 |
| $p$ | <0.0001 | <0.0001 | <0.0001 | 0.0005 | 0.0255 |

MF: mineral fertilizer; PGPR: plant-growth-promoting rhizobacteria; AMF: arbuscular mycorrhizal fungi. Different letters within a column indicate significant differences.

**Table 7.** Effects of the bio-fertilizers on leaf micronutrient concentrations (mg kg$^{-1}$).

| Treatments | Fe | Mn | Zn | Cu |
|---|---|---|---|---|
| 100%MF | 71.48 c | 23.30 e | 41.41 cd | 8.36 b |
| 60%MF + PGPR | 82.65 bc | 43.27 de | 46.88 c | 10.75 a |
| 60%MF + AMF | 171.67 a | 86.04 ab | 39.69 d | 10.97 a |
| 60%MF + PGPR + AMF | 174.81 a | 75.40 abc | 43.44 cd | 10.72 a |
| 80%MF + PGPR | 154.25 a | 92.04 a | 55.31 b | 11.94 a |
| 80%MF + AMF | 147.06 a | 64.70 bcd | 58.44 ab | 12.25 a |
| 80%MF+ PGPR + AMF | 107.63 b | 54.22 cd | 61.56 a | 11.25 a |
| $LSD_{0.05}$ | 31.121 | 25.501 | 5.598 | 1.768 |
| $p$ | <0.0001 | 0.0002 | <0.0001 | 0.0060 |

MF: mineral fertilizer; PGPR: Plant-growth-promoting rhizobacteria; AMF: arbuscular mycorrhizal fungi. Different letters within a column indicate significant differences.

Baum et al. [29], reported that the arbuscular mycorrhizal fungi with plant-growth-promoting bacteria, with a 50% reduction of P fertilizer during seedling transplanting, increased the growth and yield of pepper plants. Thus, biofertilizers could substitute P fertilizer in pepper cultivation. Ortas [35] showed that mycorrhizal application enhanced pepper plants' P and Zn content. Bio-fertilizers enrich the root zone with plant nutrients through N fixation, P, and K mineralization and stimulate plant growth regulators [24]. In chili pepper cultivation, mycorrhiza and bacteria have reported excellent synergistic effects on pepper plant nutrition by providing significant advantages to the uptake of P, Zn, Cu, Mn, and Fe nutrients [36]. In addition, some studies [37–39] noted that co-inoculation of mycorrhiza and bacteria improves nutrient uptake. Although the mechanism is not well known, Bharadwaj et al. [40] stated that the AMF secrete carbohydrates, amino acids, and unidentified compounds that could make the environment favorable for the growth of AMF-associated bacterial growth.

## 4. Conclusions

Bacteria and mycorrhiza reduced the need for mineral fertilizers used in soilless-grown capia pepper by 20%. Furthermore, combining mycorrhiza and bacteria was more effective than their individual use. Thus, we observed a synergistic effect between the two biofertilizers. The application of the 80% MF + PGPR + AMF induced the highest yield of 5.80 kg m$^{-2}$, which is 32.4% higher than the 100% MF control yield (4.38 kg m$^{-2}$). The second- and third-highest yields were 24.2% and 11.2% higher than that of the control for the 80% MF + PGPR and 80% MF + AMF treatments, respectively. The results obtained from the study showed that the use of biofertilizers increased the yield. Therefore, using bacteria and mycorrhiza in soilless-grown capia pepper is an eco-friendly approach to a sustainable environment that reduces synthetic mineral fertilizers and protects the environment.

**Supplementary Materials:** The following supporting information can be downloaded at: https://www.mdpi.com/article/10.3390/horticulturae9020188/s1, Figure S1: Views of the capia red peppers on the plants in greenhouse (a,b) and harvested in the lab for fruit analysis (c).

**Author Contributions:** All the authors contributed to this research. H.Y.D. and M.Y. designed the experiment. Conceptualization; data curation; formal analysis; investigation; resources; funding acquisition: H.Y.D., M.Y., S.D., B.I. Supervision; writing—review, and editing: H.Y.D. and N.S.G. All authors have read and agreed to the published version of the manuscript.

**Funding:** This work was supported by the Cukurova University Research Foundation (BAP) under project number FYL-2018-10403.

**Data Availability Statement:** The data presented in this study are available in the article.

**Acknowledgments:** We thank the Research Foundation Office of Cukurova University (BAP).

**Conflicts of Interest:** The authors declare no conflict of interest.

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
