# Peer review of "Bio-Fertilizers Reduced the Need for Mineral Fertilizers in Soilless-Grown Capia Pepper"

_horticulturae, doi:10.3390/horticulturae9020188_

Round 1
Reviewer 1 Report
Specific cmments:
1. The description of Capia pepper is too insufficient, please describe in detail the biological characteristics and the reasons why the author chose this plant.
2. The author is asked to highlight the advantages of using biofertilizers by explaining how mineral fertilizers can be harmful to the environment or to crops.
3. The authors are requested to add the possibility of replication of AMF and PGPR in the method mentioned in this experiment "Reduction of intensive use of mineral fertilizers by using beneficial microorganisms such as AMF and PGPR" in terms of their costs and benefits, respectively.
4. Please explain why the author chose coco pith slabs as a soilless medium. and if coco pith slabs themselves will have an impact on crop growth and yield.
5. Please describe in detail the method by which the liquid bacteria "Medbio" was applied to the plant roots in the experiment. Explain whether the method used may have led to uneven application or other causes of error.
6. Please explain whether there is a reaction between MF, AMF and PGPR mentioned in the article, and if not, please cite the existing experimental findings to support it.
7. The experimental pepper varieties are introduced in lines 67-73, but the growing environment of pepper varieties is not introduced, and the soilless cultivation habits and suitable pH value of pepper are not described.
8. 95-97 lines of content, inoculation of mycorrhizae in the process of pepper transplantation, is it in a sterile environment? Is the influence of other microorganisms excluded?
9. What is the basis for setting pH and EC value of nutrient solution in 2.2? According to the growth habit of pepper?
10. No explanation was given for the smaller stem diameter and reduced number of branches in the 80% MF+B+M experimental group compared to the 100% MF experimental group in Table 2.
11. The data in Table 3 show that the 80% MF+B+M experimental group had more reduction in the fresh weight of buds and leaves compared to the 100% MF experimental group, and the authors did not analyze the reasons for this.
12. Figure 2 shows that the leaf area of the 80% MF+B+M experimental group is smaller than that of the 100% MF experimental group, which is not pointed out in the discussion in rows 180-187. In addition, several indicators show that the 80% MF+B+M experimental group is less effective than the 100% MF experimental group, which does not explain well the significance of this experiment.
13. 201 lines and 252 lines show text errors, %80→80%; 206 lines, %60→60%; 313 lines AMF followed by "-"
14. It can be seen in Figure 4 that 80% MF+PGPR experimental group has the best effect and 80% MF+AMF experimental group has similar effect to the control group, which shows that biofertilizer induces more plant nutrition and may also depend on the added bacteria and mycorrhizae, but it is not stated in the content of lines 220-224, but only a general description that biofertilizer has good effect of inducing plant growth.
15. Since the tested species belonged to food, the authors neglected to test the taste of the experimental peppers and whether there was a change in the taste of the peppers between different experimental groups, such as pepper spiciness, texture, etc.
16. In the conclusion section, the authors pointed out that the use of different bacteria and mycorrhizae in capia peppers grown without soil has good synergistic effect, but according to the experimental data in the article the effect achieved by adding bacteria and mycorrhizae is not much different from 100% MF, then the feasibility of the study should be considered according to the actual situation, such as economic benefits.
Reviewer 2 Report
The current study reports an interesting topic that points out the possibility of reducing mineral fertilizers and partial substituting with bio-fertilizers in soilless culture of capia pepper.
The manuscript needs some minor adjustments:
In title “caipa” please replace with “capia”
Keywords and Abstract are appropriately addressed; In lines 13 and 15 the personal mode should be avoided and passive voice should be used.
Introduction is well structured and clear. The study's objectives and problematic are clear. However, some data reporting the intensive fertilization of capia pepper would be of interest in order to sustain the study’s aim.
Lines 78-79: How “bio-fertilizers effects in soilless-grown capia pepper on plant growth, yield and fruit quality were investigated“ if a treatment with 100% bio-fertilizers was not included?
Materials and methods is well structured but needs some explanations:
Which was the seedling age?
Which was the cultivation period?
All used methods (e.g. determination of plant growth parameters; determination of nitrogen and phosphorus) should be described in a brief manner.
Results are convincing and well supported by adequate statistical analysis. Some recommendation:
In all figures it should be specified what the bars mean.
There are some inconsistencies in the name of the applied treatments (e.g. line 143 “60% MF+B+M” while the same treatment in table 2 is presented as “60%MF+PGPR+AMF”)
Line 143: “60% MF+B+M” should be changed with “60% MF+PGPR+AMF”
Line 200: “80% MF+B” should be changed with “80% MF+PGPR”
Line 201: “%80 MF+M” should be changed with “80% MF+AMF”
Line 206: “%60 MF+M” should be changed with “60% MF+ AMF”
Line 252: “%80 mineral fertilizers with mycorrhiza” should be changed with “80% mineral fertilizers with mycorrhiza”
Line 258: “80% MF+B+M” should be changed with “80% MF+PGPR+AMF”
Conclusion is clear and well aiming.
The references used are relevant and up-to-date.
